# Psychological distress and coping mechanisms due to the COVID-19 pandemic among the adult population in Bo Sierra Leone. A cross-sectional study

Peter Bai James[1,2☽], Augustus Osborne[iD][3☽]*, Fatmata Seray Bah[4☽], Abdulai Jawo Bah[2,5], Jia Bainga Kangbai[4,6], George A. Yendewa[7,8,9]

1 National Centre for Naturopathic Medicine, Faculty of Health, Southern Cross University, Lismore, Australia, 2 Faculty of Pharmaceutical Sciences, College of Medicine and Allied Health Sciences, University of Sierra Leone, Freetown, Sierra Leone, 3 Department of Biological Sciences, School of Basic Sciences, Njala University, PMB, Freetown, Sierra Leone, 4 Department of Environmental Health, School of Community Health Sciences, Njala University, PMB, Freetown, Sierra Leone, 5 Institute for Global Health and Development, Queen Margaret University Edinburg, Musselburgh, Scotland, United Kingdom, 6 Department of Public Health, Eastern Technical University of Sierra Leone, Kenema, Sierra Leone, 7 Department of Medicine, Case Western Reserve University School of Medicine, Cleveland, Ohio, United States of America, 8 Division of Infectious Diseases and HIV Medicine, University Hospitals Cleveland Medical Center, Cleveland, Ohio, United States of America, 9 Johns Hopkins Bloomberg School of Public Health, Baltimore, Maryland, United States of America

☽ These authors contributed equally to this work.
* augustusosborne2@gmail.com

**Data Availability Statement:** The dataset used and analysed during the current study is available as a supplementary file.

## Abstract

Psychological distress is widely recognized as a significant health concern that poses a potential risk to the overall mental wellbeing of individuals. This study investigated the psychological distress associated with the COVID-19 pandemic and the coping methods employed by adults in Bo district, Sierra Leone. This research used a snapshot approach (cross-sectional design) to describe the prevalence of psychological distress during the COVID-19 pandemic among 502 adults residing in Bo district, Sierra Leone. We collected study data using a structured questionnaire that comprised of participant's demographics, Kessler Psychological Distress Scale (K10), Fear of COVID-19Scale (FCV-19S), Brief Resilient Coping Scale (BRCS), and Duke-UNC Functional Social Support instrument. We used backward stepwise binary logistic regression to identify the key factors linked to psychological distress. The average psychological stress score was 22.96±11.35, with approximately one-third of participants (n = 160, 31.9%) exhibiting very high levels of psychological distress. The mean score for fear of COVID-19 was 29.71±6.84, with the majority (n = 420,83.7%) being fearful of COVID-19. The mean score for the Brief Resilient Coping Mechanism was 12.49±4.51, with half of the participants considered low resilient copers (n = 257,51.2%). The mean score for functional social support was 25.35±8.85, with (n = 240, 47.8%) having increased social support. Individuals with a known health condition [aOR = 4.415, 95% CI = 1.859–10.484], who provided care to a family member/patient with known/ suspected COVID-19 [aOR = 4.485, 95% CI = 1.575–12.775], who knew someone who died from COVID-19 [aOR = 3.117, 95% CI = 1.579–6.154], with an increased fear of

**Funding:** The authors received no specific funding for this work.

**Competing interests:** The authors have declared that no competing interests exist.

COVID-19 [aOR = 4.344, 95% CI = 2.199–8.580] had higher odds of moderate to severe psychological distress. Moderate resilient copers [aOR = 0.523, 95% CI = 0.296–0.925] had lower odds of psychological distress compared to the low resilient copers. Participants with increased social support had lower odds of psychological distress than those with low support [aOR = 0.253, 95% CI = 0.147–0.434]. A significant proportion of the study cohort residing in Bo, Sierra Leone, experienced high levels of mental stress because of the COVID-19 pandemic. This study shows the importance of designing and implementing programs that minimize COVID-19 stressors and enhance the coping skills and social support network.

## Introduction

The emergence of COVID-19 in December 2019 caused by the SARS-COV-2 virus is considered a global public health threat after it was declared a pandemic in March 2020 [1]. Since COVID-19 was declared a pandemic, the morbidity and mortality figures have increased daily. As of 17th September 2023, 770,778 396 people worldwide have been infected, and 6,958,499 have died from COVID-19 [2]. Africa has been the least affected region, with over 9 million cases and 120,000 deaths reported so far, although its socio-economic impact, is expected to be huge and devastating in the coming years [3]. A lot of African economies rely on minerals and agricultural products as their primary source of foreign exchange. The global economic decline due to COVID-19 has decreased demand for these commodities and disrupted the supply chain for essential commodities, leading to scarcity and price hikes, negatively impacting most African economies [3, 4]. For Instance, the United Nations Development Programme (UNDP) reported reductions in GDP growth range from -2.6% to -10.6% across ten African countries in addition to a decrease in household income, reduced government revenues and increases in extreme poverty [5]. The same UNDP report projected that indirect mortality from COVID-19 will constitute 80% of under-five mortalities in 2025 and 2030 [5].

Similar to other countries in Africa, Sierra Leone has been affected equally. The first case of COVID-19 was reported on the 31st of March 2020 [6], and as of the 13th of August 2021, 6,332 and 121 confirmed cases and deaths have been reported [7]. To prevent the spread of COVID-19 in the community, the government of Sierra Leone, in the early stages of the pandemic instituted public health preventive measures. These include a temporary 9 pm to 5 am curfew, -inter-district lockdown, mandatory 14-day quarantine for international travellers, isolation overseas travel for all government officials and advice to the general public not to travel, school closure, facemask wearing and routine surveillance and monitoring of confirmed COVID-19 cases [8]. As the number of COVID-19 cases significantly decreased in the later stages (late 2022- early 2023), most of these measures were stopped although public health surveillance and vaccination continues [9]. As at October 2023, close two-third of the Sierra Leonean population have received at least one dose of COVID-19 vaccine [9].

These restrictions are believed to have had a significant impact on the social, psychological, and economic wellbeing of Sierra Leoneans [10]. COVID-19 and the restrictions put forward by the government to prevent it from spreading in the community may lead to fear contagion, unemployment, family separations, domestic violence, deaths of loved ones, social stigmatization, increased loneliness, work stress, the overabundance of (mis)information on social media [11]. All these issues are potential stressors and risk factors for mental health disorders such as anxiety, depression, and post-traumatic stress disorder [12, 13]. These stressors may have been compounded by the fact that the general population may still suffer from past trauma due to the 2013–2016 Ebola outbreak [14, 15].

It has been documented that when people experience infectious disease outbreaks and other traumatic events, they exhibit varied responses based on their coping styles, social support, and perceived psychological distress levels [15]. Coping mechanisms refer to behavioural and psychological actions used to tolerate, reduce, or minimize stressors due to a traumatic event [16]. In response to the trauma caused by recent infectious disease outbreaks, people have developed a variety of coping mechanisms, which can be positive or negative [15]. Positive coping strategies lead to positive mental health outcomes, while negative coping mechanisms can further exacerbate existing mental health pathologies [17].

Several studies worldwide have examined the psychological impact of COVID-19 among healthcare workers [18–20] and the general population16 [21]. Studies conducted in China, the USA, the UK, Australia, and Ethiopia have reported high levels of COVID-19-related psychological distress among the general population [11, 20–24]. Similarly, studies have reported various positive (problem-solving and positive reappraisal and social support and negative (Alcohol and cigarette consumption) coping strategies that people have employed to cope with COVID-19-related stressors [25–27]. Also, higher psychological distress is correlated with negative coping strategies [22]. In Sierra Leone, COVID-19 received unprecedented media attention, despite low risk of transmission due to the relative low number of confirmed cases and deaths [28]. Also, previous trauma due Ebola, the government COVID-19 response initiatives and sensationalized media coverage of the COVID-19 may have contributed to widespread fear, worry and disproportionate behaviour among the general Sierra Leonean population [29]. Food insecurity due to economic impacts of COVID-19 and exposure to COVID-19 are known to have contributed to people anxiety [30]. Given that Sierra Leoneans have experienced trauma due to Ebola [31, 32], and studies in other Ebola -affected countries have reported that the general population were psychologically affected years after the end of the outbreak [33, 34], we wanted to know to what extent has recent COVID-19 outbreak impacted the general population.

In Sierra Leone, the psychological impact of COVID-19 is not well known. Studies among disabled and disadvantaged children and young people aged 12 to 25 years as well as journalists reported relatively poor mental health [35]. A phone-based survey from Mozambique, Sierra Leone, Tanzania and Uganda on COVID-19- related anxiety reported a 17.0% prevalence in Sierra Leone [30]. Other COVID-19 related studies have explored healthcare system preparedness [36, 37], COVID-19 related knowledge, attitudes and practices [37, 38], SAR-COV-2 seroprevalence [39] and COVID-19 management [40]. Data on the elements linked to psychological distress in Sierra Leone due to COVID-19 outbreak are scarce. As a result, it is critical to comprehend the scope of the mental health burden in Sierra Leonean communities during the COVID-19 pandemic. This research addresses this knowledge gap by determining the level of psychological distress and its associated factors among adults ($\geq$ 18 years) in Bo district, Southern Sierra Leone.

## Methods

### Study design

A community cross-sectional survey among adults ($\geq$ 18 years) was conducted between June and July 2023 in Bo city, Southern Sierra Leone.

### Study setting

The research was conducted within the Bo district in the Southern province of Sierra Leone. The district ranks fifth most populated in the nation, encompassing 15 rural chiefdoms and 24 urban regions [41]. The district headquarters of this region, Bo, holds the distinction of being

the second most populous city in Sierra Leone [41]. The district's population is officially documented as 575,478, representing 8.1% of the country's total population. Two-thirds of the district's residents reside in rural regions [41]. Bo district was selected because it recorded the fourth largest confirmed COVID-19 cases, according to the Directorate of Health Security and Emergencies Ministry of Health and Sanitation, Government of Sierra Leone [42]. Western Urban (Freetown) recorded 462 cases, followed by PortLoko, Kenema and Bo districts, which had 231, 173 and 149 cases respectively [42].

## Study population and sampling

A sample of 508 individuals from the general community of Bo district, Sierra Leone, was recruited for the study. The current study included adults male and female individuals from Bo Sierra Leone who met the eligibility criteria of being 18 years old and have lived in the study area for at least six months and expressed a willingness to participate. The 2015 census stated that 298,623 of the population in Bo district are 18 years and above [41]. Sample size calculation for this study was conducted using the formula for calculating sample size for a cross-sectional study. The formula is as follows: $n = Z2\ Pq\ /\ d2$, where n represents the sample size, d represents the degree of accuracy or standard error, q = 1-P, Z represents the value of the test statistics (which is 1.96), and P represents the estimated proportion of prevalence of psychological distress. A percentage value of 47% was employed, denoted as 0.47 or P, drawing inspiration from similar research carried out in Ethiopia [43]. After the final calculation below, the sample size was rounded to 508 respondents to compensate for non-response. $n = Z2\ pq/d2$ the $n = (3.8416\ x0.47\ x0.53)/0.0025$. $n = 384$.

Of the 15 chiefdoms that constitute Bo district [41], only Bo city is urban. We estimated 200 respondents were needed from Bo city and 308 from the remaining 14 rural chiefdoms. We divided Bo City into 20 communities and randomly selected five from a randomly ordered list. In each of the five communities, we randomly chose 40 participants. Also, we recruited 22 participants in each of the 14 rural chiefdoms. In each selected neighbourhood in Bo City, data collectors randomly chose a starting point, and every fifth household was chosen while walking along street. At most, two adults were allowed to be recruited and interviewed from each household. Adults from each household were selected based on availability during the interview.

## Instruments

**Psychological distress scale.** We used the Kessler Psychological distress (K10) to measure levels of psychological distress [44, 45]. The K10 scale included ten questions that measure emotional distress by assessing anxiety and depressive symptoms. K10 is a valid and reliable tool with wide applications in research and clinical practice in African countries [46–48]. Participants were asked to respond to the Kessler Psychological distress Scale using a 5-point Likert scale, ranging from 'none' to 'all the time,' indicating how frequently they experienced each symptom. The psychological Distress Scale showed a very good internal consistency (Cronbach's alpha 0.978) in our study. All ten items were scored, and the total score was categorised into four levels of psychological distress: low (10–15), moderate (16–21), high (22–29), and very high (30–50). Previous studies informed our categorisation of psychological distress due to COVID-19 [11, 43].

**Level of fear of COVID-19.** Fear of COVID-19 was assessed using a recently validated Fear of COVID-19 scale (FCV-19S) [49]. FCV-19S has seven questions, and each question used a 5-point Likert scale, where people could choose how much they agreed with statements about COVID-19. The options ranged from 'strongly disagree' to 'strongly agree. The

questions in the fear of COVID-19 scale effectively measured the same concept (fear of COVID-19) in our study (Cronbach's alpha = 0 .838). We scored all seven items, and the total score was categorised into low (score 7–21) and high (score 22–35) [11].

**Coping scale.**    To assess coping skills, we used the Brief Resilient Coping Scale (BRCS) [50, 51]. It is a 4-item measure that has been used in various studies [7, 31, 32]. The BRCS showed excellent internal consistency (Cronbach's alpha 0.973) in our study. All four items were scored, and the total score was used to categorize participants into three groups based on their coping tendencies. Those with scores of 4–13 were classified as 'low resilient copers,' scores of 14–16 indicated 'medium resilient copers,' and scores of 17–20 meant 'high resilient copers' [11].

**Social support.**    We measured social support using the Duke-UNC Functional Social Support Questionnaire (FSSQ) [52]. This is an eight-item instrument to measure the strength of a person's perception and need for a social support network. It has been used to measure social support during COVID-1934. The Duke-UNC Functional Social Support scale showed an excellent internal consistency (Cronbach's alpha 0. 978) in our study. The eight-item FSSQ scale was scored, and the total score was defined into low social support ($\leq$27) and high social support ($>$27). Given that there is no recommended cutoff point for FSSQ, we chose the median of the total scores in our study as the cutoff point [53, 54].

**Data collection.**    Trained data collectors administered the questionnaire to a selected household member. The questionnaire was self-administered, or interviewer administered. The questionnaire consists of sociodemographic and health-related questions and the three scales described above.

**Data analysis.**    Data was analysed using SPSSv28. Descriptive statistics was represented in frequency and percentages, mean and standard deviation. K10 was categorised into low (score 10–15) and moderate to very high (score 16–50), fear of covid was categorised as low (score 7–21) and high (score 22–35), BRCS was categorised into participants with low (score4–13), medium r (score 14–16) and high (score 17–20) resilience in coping with Psychological stress due to COVID-19 [11], and FSSQ was defined into low Social support ($\leq$27) and high social support ($>$27). The chi-square test assessed the relationship between sociodemographic, health-related, and psychological stress levels. Binary logistic regression was employed to determine the magnitude and significance of associations between Psychological distress and independent variables. We conducted a univariate analysis for all variables to get a sense of their distribution. We employed backward stepwise binary logistic regression to create the most concise model that identifies the key sociodemographic and health-related factors predicting moderate to very high psychological distress.

**Ethical considerations.**    Our research was granted ethical clearance by the Sierra Leone Ethics and Scientific Review Committee [SLESRC No: 015/02/2023]. Each participant was given a participant information sheet explaining the nature and scope of the study. Participants were informed that they could opt out of the study at any time. A written informed consent form was given to participants in this study. Signing or thumb-printing the consent form will be considered willingness to participate in the study. To ensure confidentiality, data collected from participants was stored in a secured file using a password-protected computer, and survey responses were anonymous, with each participant given an identification number.

## Results

Out of the 510 that were approached and filled the questionnaire, a total of 502 individuals were included in our analysis. Table 1 provides details of the background information of study participants. Approximately a quarter (n = 131,26.1%) are in the 18–27 age group. Close to

half of the respondents were male (n = 229, 45.6%), married (n = 237, 47.2%) and identified as Christians. (n = 226, 45.0%). Most participants were from the Mende ethnic group (n = 405,80.7%). A quarter of them have had secondary education (n = 126,25.1%). A large majority (over two-thirds) of the study participants were unemployed (n = 348,69.3%). Approximately a fifth (n = 93,18.5%) of participants had visited a healthcare facility. Most participants (n = 412,82.1%) reported that COVID-19 had impacted their financial situation. Close to two-thirds (n = 305,60.8%) of participants reported that their households do not have a car, television, computer, or refrigerator. Close to a quarter of participants were COVID-19 frontline workers (n = 111,22.1%). Most participants (n = 392,78.1%) had never smoked, while a fifth reported drinking alcohol in the last four weeks (n = 105, 20.9%). Very few (n = 5,1.0%) had been diagnosed with COVID-19. Approximately one in ten (n = 46, 9.2%) of study participants had cared for someone in their family or patient with a known or suspected case of COVID-19, Close to a third (n = 152,30.3%) participants knew someone who had died from COVID-19, while close to two-thirds (n = 307, 61.3%) participants disagreed that COVID-19 is no longer a threat in Sierra Leone.

Table 2 shows the psychological distress level among the study participants, which was measured using the Anxiety and Depression Checklist (K10) over the last four weeks. The checklist consists of several questions about the frequency of experiencing specific symptoms of psychological distress, such as feeling tired out for no good reason, feeling nervous, hopeless, restless, or fidgety, feeling depressed, and feeling worthless. The responses were recorded on a scale ranging from "none of the time" to "all of the time." The mean score for psychological distress was 22.96±11.35. The total score on the questionnaire determined a participant's level of psychological distress. Scores ranged from 10 to 50, with higher scores indicating greater distress. We categorized the participants into four groups: mild distress (10–15), moderate distress (16–21), high distress (22–29), and very high distress (30–50). One-third of participants (n = 160, 31.9%) fell into the "very high" category, indicating a significant level of psychological distress.

Table 3 shows the fear of COVID-19 among the study participants (N = 502) based on a seven-item scale called the Fear of COVID-19 Scale. The participants' mean score was 29.71 ±6.84, with a low score of 7–21 and a high score of 22–35. The results show that the majority of participants strongly agree that they are most afraid of COVID-19(n = 470,93.6%), uncomfortable thinking about it(n = 429,85.5%), and afraid of losing their life because of COVID-19 (n = 460,91.6%). Additionally, nearly two-thirds of participants strongly agree that they became nervous or anxious when watching news and stories about COVID-19 on social media (n = 307,61.2%).

Table 4 shows the Brief Resilient Coping Mechanism among the 502 participants. The study participants were asked to rate their agreement with statements related to four coping mechanisms statements: "looking for creative ways to alter difficult situations", "believing they can control their reaction to what happens to them", "thinking they can grow in positive ways by dealing with difficult situations", and "actively looking for ways to replace the losses they encounter in life". The participants indicated their level of agreement with each statement on a Likert scale ranging from 'Does not describe me at all' to 'Describes me very well'. The results show that the mean score for the Brief Resilient Coping Mechanism was 12.49±4.51, with half of the participants considered as low resilient copers (n = 257,51.2%). The study found that most participants described themselves as having the coping mechanisms described in the statements, with the highest percentage of participants (n = 219, 43.6%) rating the statement "I look for creative ways to alter difficult situations" as describing them.

**Table 1. Characteristics of the study population(N = 502).**

| Characteristics | Variable | n (%) |
|---|---|---|
| Age | 18–27 | 131(26.1) |
| | 28–37 | 126(25.1) |
| | 38–47 | 97(19.3) |
| | 48–57 | 75(14.9) |
| | 58 and above | 73(14.5) |
| Sex | Male | 229(45.6) |
| | Female | 273(54.4) |
| Marital Status | Single | 153(30.5) |
| | Married | 237(47.2) |
| | Divorced | 7(1.4) |
| | Widowed | 43(8.6) |
| | Cohabiting | 62(12.4) |
| Religion | Christianity | 226(45.0) |
| | Islam | 276(55.0) |
| Ethnicity | Mende | 405(80.7) |
| | Temne | 45(9.0) |
| | Krio | 5(1.0) |
| | Limba | 11(2.2) |
| | others | 36(7.2) |
| Level of Education | Non-Formal | 189(37.6) |
| | Primary | 30(6.0) |
| | Secondary | 126(25.1) |
| | Tertiary | 157(31.3) |
| Employment Status | Unemployed | 348(69.3) |
| | Employed | 154(30.7) |
| Visited a healthcare facility (hospital, clinic) in the past 4weeks | Yes | 93(18.5) |
| | No | 409(81.5) |
| COVID-19 impacted the financial situation | Yes | 412(82.1) |
| | No | 90(17.9) |
| Does your household have any of the following items? (Car, Television, computer, refrigerator) | Yes | 197(39.2) |
| | No | 305(60.8) |
| Are you a COVID-19 frontline worker (Healthcare worker, contact tracer, etc.) | Yes | 111(22.1) |
| | No | 391(77.9) |
| Known health condition | Yes | 83(16.5) |
| | No | 419(83.5) |
| Smoking status | Never Smoked | 392(78.1) |
| | Ever smoked | 48(9.6) |
| | currently smoking | 62(12.4) |
| Current alcohol drinking (last four weeks) | Yes | 105(20.9) |
| | No | 397(79.1) |
| Ever diagnosed with COVID-19 | Yes | 5(1.0) |
| | No | 497(99.0) |
| Provided care to a family member/patient with a known/suspected case of COVID-19 | Yes | 46(9.2) |
| | No | 456(90.8) |
| Know someone quarantined for COVID-19: | Yes | 201(60.0) |
| | No | 301(40.0) |

(*Continued*)

**Table 1.** (Continued)

| Characteristics | Variable | n (%) |
|---|---|---|
| Know someone who died from COVID-19. | Yes | 152(30.3) |
| | No | 350(69.7) |
| COVID-19 is no longer a threat in Sierra Leone | Disagree | 307(61.3) |
| | Agree | 195(38.8) |

Table 5 shows the level of functional social support among the study participants (N = 502). The participants were asked to rate their level of social support using the following statements "having people who care about them", "getting love and affection", "chances to talk to someone about problems", "getting useful advice", and "help when sick in bed". The responses were rated on a scale from "much less than I would like" to "as much as I would like." The mean score for functional social support was 25.35±8.85, and participants were categorized as having low social support (≤27) or high social support (>27). Of the 502 participants, (n = 262,52.2%) had low social support, while half (n = 240,47.8%) had high social support. Half of the participants agreed to the statement that they have people who care what happens to them at least as much as they would like (n = 258, 51.3%)

Table 6 presents the results of the regression analysis examining factors associated with high K10 scores (indicating psychological distress) in the study population. In the unadjusted analysis, the age group 28–47 years had higher odds of moderate to very high psychological distress than the age group of 18–27. However, this association was not significant in the adjusted analysis. The age group 48 and above also had higher odds of psychological distress, but this association was insignificant in both the unadjusted and adjusted analyses. There was no significant difference in the odds of psychological distress between males and females in both the unadjusted and adjusted analyses. The married/cohabitating group had higher odds of psychological distress than the single/divorced/widowed in the unadjusted analysis. However, this association was not significant in the adjusted analysis. The unemployed individuals had higher odds of psychological distress compared to those employed in the unadjusted analyses. However, this association was not significant in the adjusted analysis. The individuals with these household items (Car, Television, computer refrigerator) had lower odds [aOR = 0.351, 95% CI = 0.200–0.617] of psychological distress compared to those with no household items in both the unadjusted and adjusted analyses. The individuals who were COVID-19 frontline workers had lower odds of psychological distress [aOR = 0.211, 95% CI = 0.100–0.444] compared to those who were not frontline workers in both the unadjusted and adjusted analyses. Individuals with a known health condition had higher odds of psychological distress [aOR = 4.415, 95% CI = 1.859–10.484] compared to those with no known health condition. Participants who cared for a family member or someone with known/suspected COVID-19 had higher odds of psychological distress [aOR = 4.485, 95% CI = 1.575–12.775] compared to those who did not provide care. Individuals who knew someone who died from COVID-19 had higher odds of psychological distress [aOR = 3.117, 95% CI = 1.579–6.154] than those who did not know someone who died from COVID-19. Individuals with significant fear of COVID-19 had higher odds of psychological distress compared [aOR = 4.344, 95% CI = 2.199–8.580] to those with low fear of COVID-19. The moderate resilient copers [aOR = 0.523, 95% CI = 0.296–0.925] and high resilient copers [aOR = 0.647, 95% CI = 0.259–1.621] had lower odds of psychological distress compared to the low resilient copers. Participants with high social support had lower odds of psychological distress [aOR = 0.253, 95% CI = 0.147–0.434] than those with inadequate support.

**Table 2. Psychological distress among the study participants(N = 502).**

| Anxiety and Depression Checklist (K10) (last four weeks) | Variable | Total, n (%) |
|---|---|---|
| About how often did you feel tired out for no good reason? | None of the time | 170(33.9) |
| | a little of the time | 98((19.5) |
| | some of the time | 129(25.7) |
| | most of the time | 71(14.1) |
| | all of the time | 34(6.8) |
| In the past four weeks, how often did you feel nervous? | None of the time | 196(39.0) |
| | a little of the time | 97(19.3) |
| | some of the time | 108(21.5) |
| | most of the time | 70(13.9) |
| | all of the time | 31(6.2) |
| In the past four weeks, how often did you feel so nervous that nothing could calm you down? | None of the time | 202(40.2) |
| | a little of the time | 117(23.3) |
| | some of the time | 93(18.5) |
| | most of the time | 61(12.2) |
| | all of the time | 29(5.8) |
| In the past four weeks, how often did you feel hopeless? | None of the time | 222(44.2) |
| | a little of the time | 115(22.9) |
| | some of the time | 78(15.5) |
| | most of the time | 60(12.0) |
| | all of the time | 27(5.4) |
| In the past four weeks, how often did you feel restless or fidgety? | None of the time | 196(39.0) |
| | a little of the time | 136(27.1) |
| | some of the time | 79(15.7) |
| | most of the time | 62(12.4) |
| | all of the time | 29(5.8) |
| In the past four weeks, how often did you feel so restless you could not sit still? | None of the time | 198(39.4) |
| | a little of the time | 135(26.9) |
| | some of the time | 82(16.3) |
| | most of the time | 60(12.0) |
| | all of the time | 27(5.4) |
| In the past four weeks, how often did you feel depressed? | None of the time | 85(16.9) |
| | a little of the time | 106(21.1) |
| | some of the time | 140(27.9) |
| | most of the time | 108(21.5) |
| | all of the time | 63(12.5) |
| In the past four weeks, how often did you feel that everything was an effort? | None of the time | 169(33.7) |
| | a little of the time | 131(26.1) |
| | some of the time | 95(18.9) |
| | most of the time | 74(14.7) |
| | all of the time | 33(6.6) |
| In the past four weeks, how often did you feel so sad that nothing could cheer you up? | None of the time | 174(34.7) |
| | a little of the time | 145(28.9) |
| | some of the time | 97(19.3) |
| | most of the time | 60(12.0) |
| | all of the time | 26(5.2) |

(*Continued*)

**Table 2.** (Continued)

| Anxiety and Depression Checklist (K10) (last four weeks) | Variable | Total, n (%) |
|---|---|---|
| In the past four weeks, how often did you feel worthless? | None of the time | 218(43.4) |
| | a little of the time | 126(25.1) |
| | some of the time | 76(15.1) |
| | most of the time | 55(11.0) |
| | all of the time | 27(5.4) |
| Psychological Distress | Mean± standard deviation | 22.96±11.35 |
| | Mild (score 10–15) | 172(34.3) |
| | Moderate (score 16–21) | 107(21.3) |
| | high (score 22–29) | 63(12.5) |
| | Very high (score 30–50) | 160(31.9) |

## Discussion

The COVID-19 pandemic resulted in significant psychological distress, affecting individuals and entire social groups. Individuals may endure varying levels of psychological crises. This study aimed to examine the extent of psychological distress and coping mechanisms employed by a representative sample of individuals from Sierra Leone in the aftermath of the COVID-19 pandemic.

The findings of this study revealed that over thirty percent of the participants exhibited severe psychological distress, similar to a study conducted by Elkayal et al. [54]. Specifically, they reported higher levels of depression, feelings of exertion, and restlessness. Conversely, lower scores were observed for feelings of hopelessness, nervousness, inability to find solace, worthlessness, and restlessness. The observed outcomes may be attributed to the government's Implementation of stringent social distancing measures in response to concerns regarding the potential transmission of the Severe Acute Respiratory Syndrome Coronavirus-2 (SARS-CoV-2) virus within the country [8]. The implemented measures encompassed the prohibition of activities involving large gatherings, the closure of educational institutions, the Implementation of stay-at-home orders, the cancellation of significant cultural and sporting events, the closure of local enterprises, and the temporary suspension of civil and religious rituals, such as funerals and weddings [8]. The procedures were implemented to reduce the transmission of the SARS-CoV-2 virus. Public officials and relevant parties urged individuals to minimise participation in gatherings and meetings [8].

The findings from our study revealed that half of participants exhibited low coping mechanisms among participants. This could be attributed to past trauma due to the eleven years old civil war and poverty which impacted people mental health. Thus, the COVID-19 pandemic may have brought back trauma-related symptoms making it difficult for people to cope [55, 56]. Also, lockdown and restrictions leading to economic hardship and fear of infection are sources of stress, which negatively impact people's mental health and their ability to cope with everyday life [57].

The findings from our study revealed a high fear of COVID-19 among participants. The unpredictability and uncertainty surrounding the disease might have created fear among people [55, 56]. The seriousness of the illness and the deaths caused by it have inflicted a sense of fear among people [58]. Information gaps about the disease may also have contributed to fear among people [58] A study by Rahman et al suggested that individuals with existing health conditions are more likely to be infected by COVID-19, and that might have contributed to

Table 3. Fear of COVID-19 among the study participants(N = 502).

| Fear of COVID-19 statements | Variables | n (%) |
|---|---|---|
| I am most afraid of COVID-19. | Strongly Disagree | 2(0.4) |
| | Disagree | 4(0.8) |
| | Neutral | 2(0.4) |
| | Agree | 24(4.8) |
| | Strongly Agree | 470(93.6) |
| It makes me uncomfortable to think about COVID-19 | Strongly Disagree | 27(5.4) |
| | Disagree | 4(0.8) |
| | Neutral | 6(1.2) |
| | Agree | 36(7.2) |
| | Strongly Agree | 429(85.5) |
| My hands become clammy when I think about COVID-19 | Strongly Disagree | 72(14.3) |
| | Disagree | 29(5.8) |
| | Neutral | 19(3.8) |
| | Agree | 32(6.4) |
| | Strongly Agree | 350(69.7) |
| I am afraid of losing my life because of COVID-19 | Strongly Disagree | 5(1.0) |
| | Disagree | 8(1.6) |
| | Neutral | 8(1.6) |
| | Agree | 21(4.2) |
| | Strongly Agree | 460(91.6) |
| When watching news and stories about COVID-19 on social media, I become nervous or anxious. | Strongly Disagree | 97(19.3) |
| | Disagree | 29(5.8) |
| | Neutral | 19(3.8) |
| | Agree | 50(10.0) |
| | Strongly Agree | 307(61.2) |
| I cannot sleep because I'm worrying about getting COVID-19 | Strongly Disagree | 113(22.5) |
| | Disagree | 40(8.0) |
| | Neutral | 22(4.4) |
| | Agree | 43(8.6) |
| | Strongly Agree | 284(56.6) |
| My heart races or palpitates when I think about getting COVID-19 | Strongly Disagree | 122(24.3) |
| | Disagree | 33(6.6) |
| | Neutral | 23(4.6) |
| | Agree | 58(11.6) |
| | Strongly Agree | 266(53.0) |
| Fear of COVID-19 | Mean± standard deviation | 29.71 ±6.84 |
| | Low (score 7–21) | 82(16.3) |
| | High (score 22–35) | 420(83.7) |

fear among them [11]. Elkayal et al. [54] also reported high levels of fear of COVID-19 among study participants.

Our study revealed that individuals with known health conditions were associated with high psychological distress. Knowing they have a health condition might make people more likely to develop health anxiety and become hypervigilant. This constant worry can stem from the fear of COVID-19 potentially causing serious illness or complications due to their underlying condition [55]. People with pre-existing health conditions may perceive themselves as

Table 4. Brief Resilient Coping Mechanism among the study participants(N = 502).

| Brief Resilient Coping Mechanism Statements | Variables | n (%) |
|---|---|---|
| I look for creative ways to alter difficult situations | Does not describe me at all | 73(14.5) |
| | Does not describe me | 80(15.9) |
| | Neutral | 98(19.5) |
| | Describe me | 219(43.6) |
| | Describe me very well | 32(6.4) |
| Regardless of what happens to me, I believe I can control my reaction to it | Does not describe me at all | 67(13.3) |
| | Does not describe me | 91(18.1) |
| | Neutral | 98(19.5) |
| | Describe me | 212(42.2) |
| | Describe me very well | 34(6.8) |
| I believe I can grow in positive ways by dealing with difficult situations | Does not describe me at all | 65(12.9) |
| | Does not describe me | 72(14.3) |
| | Neutral | 113(22.5) |
| | Describe me | 214(42.6) |
| | Describe me very well | 38(7.6) |
| I actively look for ways to replace the losses I encounter in life | Does not describe me at all | 66(13.1) |
| | Does not describe me | 80(15.9) |
| | Neutral | 127(25.3) |
| | Describe me | 197(39.2) |
| | Describe me very well | 32(6.4) |
| Brief Resilient Coping Mechanism | Mean± standard deviation | 12.49±4.51 |
| | Low (score4–13) | 257(51.2) |
| | Medium (score 14–16) | 195(38.8) |
| | High (score 17–20) | 50(10.0) |

more vulnerable to the virus, which can result in heightened stress and anxiety. The pandemic led to disruptions in healthcare systems, with non-COVID medical treatments and appointments being postponed or cancelled [55]. This disruption can cause additional distress for individuals who rely on regular medical care for their pre-existing conditions. In a similar study by Rahman et al. [11], a high association level was found between individuals with known health conditions and increased psychological distress.

Our study revealed that individuals who provided care to family members/patients with known suspected cases of COVID-19 were associated with high psychological distress. Possible explanations for this are that caregivers were often in close contact with COVID-19 patients, increasing their risk of infection. Fear of contracting or spreading the virus to others can lead to significant distress and anxiety [58]. In some cases, caregivers faced shortages of PPE, making it difficult to protect themselves and others adequately [37]. During the early stages of the pandemic, there was much uncertainty about the virus, its transmission, and how to protect oneself and others. Thus, caregivers may have felt overwhelmed by the constantly changing information and guidelines. In a similar study by Rahman et al. [11], a high association level was also found between individuals providing care to family or patients with known or suspected COVID-19 and increased psychological distress.

Our study revealed that individuals who knew someone who died from Covid-19 were associated with high psychological distress. Grieving individuals may experience a range of emotions, including sadness, anger, guilt, and helplessness, all contributing to psychological distress [59]. The inability to say goodbye or prepare for the loss can intensify feelings of

**Table 5. Functional Social Support among the study participants(N = 502).**

| Functional Social Support statements | Variables | n (%) |
|---|---|---|
| I have people who care what happens to me. | Much less than I would like | 62(12.4) |
| | Less than I would like | 69(13.7) |
| | Some, but I would like more | 113(22.5) |
| | Almost as much as I would like | 194(38.6) |
| | As much as I would like | 64(12.7) |
| I get love and affection. | Much less than I would like | 64(12.7) |
| | Less than I would like | 73(14.5) |
| | Some, but I would like more | 119(23.7) |
| | Almost as much as I would like | 188(37.5) |
| | As much as I would like | 59(11.6) |
| I get chances to talk to someone about problems at work or with my housework. | Much less than I would like | 57(11.4) |
| | Less than I would like | 88(17.5) |
| | Some, but I would like more | 125(24.9) |
| | Almost as much as I would like | 181(36.1) |
| | As much as I would like | 51(10.2) |
| I get chances to talk to someone I trust about my personal or family problems. | Much less than I would like | 58(11.6) |
| | Less than I would like | 90(17.9) |
| | Some, but I would like more | 135(26.9) |
| | Almost as much as I would like | 169(33.7) |
| | As much as I would like | 50(10.0) |
| I get chances to talk about money matters. | Much less than I would like | 58(11.6) |
| | Less than I would like | 102(20.3) |
| | Some, but I would like more | 132(26.3) |
| | Almost as much as I would like | 158(31.5) |
| | As much as I would like | 52(10.4) |
| I get invitations to go out and do things with other people | Much less than I would like | 62(12.4) |
| | Less than I would like | 89(17.7) |
| | Some, but I would like more | 134(26.7) |
| | Almost as much as I would like | 166(33.1) |
| | As much as I would like | 51(10.2) |
| I get useful advice about important things in life | Much less than I would like | 59(11.8) |
| | Less than I would like | 89(17.7) |
| | Some, but I would like more | 136(27.1) |
| | Almost as much as I would like | 165(32.9) |
| | As much as I would like | 53(10.6) |
| I get help when I am sick in bed. | Much less than I would like | 62(12.4) |
| | Less than I would like | 70(13.9) |
| | Some, but I would like more | 103(20.5) |
| | Almost as much as I would like | 199(39.6) |
| | As much as I would like | 68(13.5) |
| Functional Social Support | Mean± standard deviation | 25.35±8.85 |
| | Low Social support (≤27) | 262(52.2) |
| | High Social Support (>27) | 240(47.8) |

distress and shock. In their study, Joaquim et al. [59] found that the experience of losing a family member or friend as a direct result of SARS-COV-2 infection exacerbates psychological distress among the individuals surveyed. In her study, Grace [60] reported that individuals who have experienced the loss of a loved one due to the virus exhibit higher levels of depressed

**Table 6. Factors associated with moderate to very high psychological distress among the study population (based on K10 score).**

| Characteristics | Variable | Moderate to Very High (Score 16–50), n (%) | Low (Score 10–15), n (%) | Unadjusted analyses | | Adjusted analyses | |
|---|---|---|---|---|---|---|---|
| | | | | OR 95% CIs | p-value | AOR (95%) CIs | p-value |
| Age | 18-27yrs | 93(28.2) | 38(22.1) | 1 | | | |
| | 28-47yrs | 126(38.2) | 97(56.4) | 0.531(0 .335–0.842) | 0 .007 | - | |
| | 48 and above | 111(33.6) | 37(21.5) | 1.226(0 .722–2.082) | 0 .451 | - | |
| Sex | Male | 152(46.1) | 77(44.8) | 1.054(0.727–1.526) | 0.782 | | |
| | Female | 178(53.9) | 95(55.2) | 1 | | - | |
| Marital Status | Single/ Divorced/ widowed | 119(36.1) | 84(48.8) | 1 | | - | |
| | married/cohabitating | 211(63.9) | 88(51.2) | 1.693(1.164–2.460) | 0.006 | - | |
| Religion | Christian | 132(40.0) | 94(54.7) | 0.553(0.381–0.803) | 0 .002 | - | |
| | Muslim | 198(60.0) | 78(45.3) | 1 | | - | |
| Ethnicity | Mende | 272(82.4) | 133(77.3) | 1.375(0.872–2.169) | 0.171 | - | |
| | Others | 58(17.6) | 39(22.7) | 1 | | - | |
| Level of Education | Non-Formal | 153(46.4) | 36(20.9) | 1 | | | |
| | Primary | 25(7.6) | 5(2.9) | 1.176(0.421–3.284) | 0.756 | – | |
| | Secondary | 88(26.7) | 38(22.1) | 0 .545(0.322–0.922) | 0.024 | - | |
| | Tertiary | 64(19.4) | 93(54.1) | 0 .162(0.100–0.262) | <0.001 | - | |
| Employment Status | Unemployed | 248(75.2) | 100(58.1) | 2.178(1.471–3.224) | <0.001 | - | |
| | Employed | 82(24.8) | 72(41.9) | 1 | | | |
| Visited a healthcare facility. | Yes | 259(79.2) | 148(86.5) | 0.592(0.354–0.990) | 0.046 | - | |
| | No | 68(20.8) | 23(13.5) | 1 | | | |
| COVID-19 impacted financial situation | Yes | 257(77.9) | 155(90.1) | 0.386(0 .220–0 .679) | <0.001 | - | |
| | No | 73(22.1) | 17(9.9) | 1 | | | |
| Does your household have any of the following items? (Car, Television, computer, refrigerator) | Yes | 82(24.8) | 115(66.9) | 0.164(0.109–0.245) | <0.001 | 0.351(0.200–0.617) | <0.001 |
| | No | 248(75.2) | 57(33.1) | 1 | | 1 | |
| Are you a COVID-19 frontline worker (Healthcare worker, contact tracer, etc | Yes | 44(13.3) | 67(39.0) | 0.241(0.155–0.375) | <0.001 | 0.211(0.100–0.444) | <0.001 |
| | No | 286(86.7) | 105(61.0) | 1 | | 1 | |
| Known health condition | Yes | 74(22.4) | 9(5.2) | 5.235(2.550–10.747) | <0.001 | 4.415(1.859–10.484) | <0.001 |
| | No | 256(77.6) | 163(94.8) | 1 | | 1 | |
| Smoking status | Never Smoked | 259(78.5) | 133(77.3) | 1 | 0.934 | - | |
| | Ever smoked | 32(9.7) | 16(9.3) | 1.027(0.544–1.939) | 0.626 | - | |
| | currently smoking | 39(11.8) | 23(13.4) | 0.871(0.499–1.518) | | - | |

(*Continued*)

**Table 6.** (Continued)

| Characteristics | Variable | Moderate to Very High (Score 16–50), n (%) | Low (Score 10–15), n (%) | Unadjusted analyses | | Adjusted analyses | |
|---|---|---|---|---|---|---|---|
| | | | | OR 95% CIs | p-value | AOR (95%) CIs | p-value |
| Current alcohol drinking (last four weeks) | Yes | 66(20.0) | 39(22.7) | 0.853(0.545–1.334) | 0.485 | - | |
| | No | 264(80.0) | 133(77.3) | 1 | | - | |
| Ever diagnosed with COVID-19 | Yes | 5(1.5%) | 0(0.0) | 1 | | - | |
| | No | 325(98.5) | 172(100.0) | 0.00(0.00–0.00) | 0.999 | - | |
| Provided care to a family member/patient with a known/suspected case of COVID-19 | Yes | 35(10.6) | 11(6.4) | 1.737(0.859–3.512) | 0.125 | 4.485(1.575–12.775) | 0.005 |
| | No | 295(89.4) | 161(93.6) | 1 | | | |
| Know someone quarantined for COVI | Yes | 130(39.4) | 71(41.3) | 0.925(0.635–1.346) | 0.682 | - | |
| | No | 200(60.6) | 101(58.7) | 1 | | - | |
| Know someone who died from COVID-19 | Yes | 115(34.8) | 37(21.5) | 1.952(1.272–2.995) | 0.002 | 3.117(1.579–6.154) | 0.001 |
| | No | 215(65.2) | 135(78.5) | 1 | | | |
| COVID-19 is no longer a threat in Sierra Leone | Disagree | 223(67.6) | 84(48.8) | 2.183(1.497–3.185) | <0.001 | - | |
| | Agree | 107(32.4) | 88(51.2) | 1 | | - | |
| Fear of COVID-19 | Low | 19(5.8) | 63(36.6) | 1 | | 1 | |
| | High | 311(94.2) | 109(63.4) | 9.461(5.418–16.520) | <0.001 | 4.344(2.199–8.580) | <0.001 |
| Brief Coping | Low resilient copers (score 4–13) | 214(64.8) | 43(25.0) | 1 | | 1 | |
| | Moderate resilient copers (score 14–16) | 94(28.5) | 101(58.7) | 0.187(0.121–0.288) | <0.001 | 0.523(0.296–0.925) | 0.026 |
| | High resilient copers (score 17–20) | 22(6.7) | 28(16.3) | 0.158(0.083–0.302) | <0.001 | 0.647(0.259–1.621) | 0.353 |
| Functional support | Low Support | 223(67.6) | 39(22.7) | 1 0.141(0.092–0.215) | <0.001 | 1 0.253(0.147–0.434) | <0.001 |

symptoms and engage in more frequent episodes of binge drinking. In their investigation, Cheng and Tang [61] found that the demise of a younger individual and the absence of a companion were associated with maladaptive consequences.

Consistent with other studies [62–64], our study indicated that participants who demonstrated moderate to high coping skills and high social support had lower odds of psychological distress compared to those with inadequate support. According to Yu et al. [22], individuals experiencing higher levels of psychological distress tend to exhibit a greater tendency toward passive coping strategies and reported lower perceived social support compared to those with lower distress. In their study, Tindle et al. [63] found that the presence of social support was associated with increased levels of psychological flexibility among the participants. In their study, Cheng et al. [64] found an inverse relationship between coping flexibility and depression and all four forms of COVID-19 anxiety. The utilisation of coping skills serves to mitigate stress and promote favourable psychological effects.

Contrary to other studies [65, 66], our research suggests that frontline workers had lower odds of psychological distress. The fact that frontline workers in Sierra Leone have previously

been exposed to a much more severe infectious outbreak like Ebola would have psychologically prepared them and increased their resilience [67]. A study by Tengbe et al. [68] reported that the Sierra Leonean health workers' increased resilience during COVID-19 was due to their prior experience in providing care to patients during the 2014 Ebola outbreak.

## Limitations

Readers should bear in mind the following study limitations when interpreting the findings of this study. First, our study findings are not representative of the adult population of the whole country, given that it was not a nationwide study. Second, given that a cross-sectional study design was used in our study, it was impossible to establish a causal linkage between the outcome and independent variables. Third, there is the possibility for recall bias as participants were asked to recall events that have happened in the past. We minimised such bias by asking participants about events that have happened recently. Fourth, 44% of participants in our study could not read and write, so an interviewer-administered format was used to collect data from this cohort. As such, there is the possibility of social desirability bias as participants reported less health impairment when interviewer-administered instruments compared to self-administered ones [69, 70]. However, other studies have reported no significant difference in patient-reported outcomes when two administration formats were used [71, 72].

It is recommended that future studies be undertaken, employing a longitudinal research approach and utilising a larger representative sample. Notwithstanding these limitations, our study is the first community-based enquiry to explore the mental health impact of COVID-19 in the Sierra Leonean population, and it would serve as a basis for future studies on the mental health impact of infectious disease outbreaks in Sierra Leone.

## Policy practice and research implications

Mental health needs of the Sierra Leonean population are high due to psychological impact of the civil war, recent Ebola and flooding and mudslide. However, such high need is mismatched by inadequate provision of mental health services [60]. The country currently has only one psychiatric hospital in the capital city, Freetown with limited workforce of two psychiatrists and nineteen mental health nurses [73]. Given the above findings, it is important that the Bo district health management team should increase equitable access to mental health services through adequate investment and integrating mental health into primary healthcare. Also, it is important that culturally appropriate mental health interventions need to be designed to meet the needs of the people. In addition, the training of healthcare providers in assessing and implementing mental health techniques is essential. The design of community mental health interventions -a way to integrate mental in in primary care is important to support people in their communities. Peer support programs have been shown to be effective community-based intervention as it provides for people to share their experiences and receive support from others who have gone through similar challenges [74]. To achieve this, multi-sectoral, multi-level coordinated effort from internal and external actors is required. Given that our study is cross-sectional in nature, it would be good to conduct longitudinal studies to monitor long-term mental health impacts of the COVID-19 to inform new or existing interventions.

## Conclusion

The present study's findings indicate that the COVID-19 pandemic resulted in psychological distress among a significant section of the general population in Bo city, Sierra Leone, with varying degrees of intensity ranging from low to high. Additionally, the present study revealed

a low coping mechanism among those experiencing psychological distress. In the current study, individuals with a known health condition had higher odds of psychological distress compared to those with no known health condition. Participants who cared for someone in their family or a patient with known/suspected COVID-19 had higher odds of psychological distress than those who did not provide care. Individuals who knew someone who died from COVID-19 were more likely to experience higher levels of psychological distress than those who did not know someone who died from COVID-19. Individuals with increased fear of COVID-19 had higher odds of psychological distress than those with low fear of COVID-19. Coping skills and social support acted as protective factors to high psychological distress. The results of our study highlight the need for effective programs to help people cope with the COVID-19 related stress. These programs could teach people skills to manage stress and build stronger social connections to better overcome the pandemic's challenges.

## Supporting information

**S1 Data. Psychological distress dataset.**
(SAV)

## Acknowledgments

We want to express our thanks and appreciation to all data collectors for your time and commitment and all research participants who agreed to take part in the study. This research is dedicated to those who have experienced adverse consequences because of the COVID-19 pandemic and to the innocent individuals who have lost their lives in Sierra Leone and the rest of the world. Also, this study is dedicated to all healthcare workers who bravely served as the first line of defence in safeguarding communities from the SARS-CoV-2 virus.

## Author Contributions

**Conceptualization:** Peter Bai James, Augustus Osborne, Fatmata Seray Bah.

**Data curation:** Peter Bai James, Augustus Osborne, Fatmata Seray Bah.

**Formal analysis:** Peter Bai James, Augustus Osborne.

**Investigation:** Peter Bai James, Augustus Osborne, Fatmata Seray Bah.

**Methodology:** Peter Bai James, Augustus Osborne, Fatmata Seray Bah, Abdulai Jawo Bah, Jia Bainga Kangbai, George A. Yendewa.

**Software:** Peter Bai James, Augustus Osborne.

**Supervision:** Peter Bai James, Augustus Osborne, George A. Yendewa.

**Validation:** Peter Bai James, Augustus Osborne, Abdulai Jawo Bah, Jia Bainga Kangbai.

**Writing – original draft:** Peter Bai James, Augustus Osborne, Fatmata Seray Bah, Abdulai Jawo Bah, Jia Bainga Kangbai, George A. Yendewa.

**Writing – review & editing:** Peter Bai James, Augustus Osborne, Fatmata Seray Bah, Abdulai Jawo Bah, Jia Bainga Kangbai, George A. Yendewa.

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
