## [Decision Letter · Decision Letter 0]

9 Aug 2024

PMEN-D-24-00222

Psychological distress and coping mechanisms due to the COVID-19 pandemic among the adult population in Bo district Sierra Leone:  cross-sectional study

PLOS Mental Health

Dear Dr. Osborne,

Thank you for submitting your manuscript to PLOS Mental Health. After careful consideration, we feel that it has merit but does not fully meet PLOS Mental Health’s publication criteria as it currently stands. Therefore, we invite you to submit a revised version of the manuscript that addresses the points raised during the review process.

Both reviewers highlight the conceptual limitations and suggest important revisions to set a firmer foundation for the study's aims and significance for publication. The reviewers also note some methodological limitations of the manuscript which requires significant revision to strengthen the conclusions. It is important to carefully consider each comment to help strengthen the manuscripts overall contribution to discourse on distress and coping during covid in a low-income context.

We look forward to receiving your revised manuscript.

Kind regards,

Lily Kpobi, Ph.D.

Academic Editor

PLOS Mental Health

Journal Requirements:

1. We noticed you have some minor occurrence of overlapping text with the following previous publication(s), which needs to be addressed:

- https://doi.org/10.1371/journal.pone.0257304

- https://doi.org/10.21203/rs.3.rs-887072/v1

- DOI:10.18203/2394-6040.ijcmph20205145

In your revision ensure you cite all your sources (including your own works), and quote or rephrase any duplicated text outside the methods section. Further consideration is dependent on these concerns being addressed.

2. We have amended your Competing Interest statement to comply with journal style. We kindly ask that you double check the statement and let us know if anything is incorrect. 

3. Please ensure that the Title in your manuscript file and the Title provided in your online submission form are the same.

4. In the online submission form, you indicated that The datasets used and analysed during the current study are available from the corresponding author upon reasonable request. 

a. In a public repository, 

b. Within the manuscript itself, or 

c. Uploaded as supplementary information.

Reviewers' comments:

Reviewer's Responses to Questions

**Comments to the Author**

1. Does this manuscript meet PLOS Mental Health’s publication criteria? Is the manuscript technically sound, and do the data support the conclusions? The manuscript must describe methodologically and ethically rigorous research with conclusions that are appropriately drawn based on the data presented.

Reviewer #1: Yes

Reviewer #2: Partly

2. Has the statistical analysis been performed appropriately and rigorously?

Reviewer #1: Yes

Reviewer #2: I don't know

3. Have the authors made all data underlying the findings in their manuscript fully available (please refer to the Data Availability Statement at the start of the manuscript PDF file)?

Reviewer #1: Yes

Reviewer #2: Yes

4. Is the manuscript presented in an intelligible fashion and written in standard English?

Reviewer #1: Yes

Reviewer #2: Yes

5. Review Comments to the Author

Reviewer #1: This is an interesting study. The manuscript provides an interesting set of findings about the psychological impact of COVID-19, which will be useful for other studies to draw upon. It can be published. However, there are some areas that could be elaborated on or improved.

In the introduction, there is a statement about the potential for the impact of COVID-19 to be huge and devastating in the coming years. Could this point be elaborated on in terms of why these negative impacts are expected and in which areas? The limitation of findings not being generalizable beyond the Bo district has been acknowledged. Nonetheless, it would be helpful to see more information on why this district was selected. If data is available, it would also be helpful to see the COVID-19 cases in the Bo district compared to other districts. A discussion of the implications of this on the findings would be useful. In the methodology section, it would be helpful to know how the interviewees from each household were selected. Was it random sampling?

Another point to consider is whether the format of survey administration may have had any impact on the responses. It was stated that some questionnaires were self-administered while others were administered by the interviewers. Why was this the case? Could percentages for each survey administration mode be provided, and could implications for interpreting the findings be stated?

Lastly, in relation to the findings, it is stated that frontline workers had lower odds of psychological distress. However, this point is not discussed in the discussion section. This point could be elaborated on in the discussion.

Reviewer #2: This paper has many strengths, but I found its underlying purpose and its contribution to the field difficult to identify. The data were collected in June/ July 2023. Vaccinations for Covid-19 began in March 2021 in Sierra Leone, and by 2023 there were no deaths from the disease (only 3 in 2022). People living in Sierra Leone have certainly experienced many challenges but it seems that by June 2023 Covid-19 wasn't one of the most significant, so it's difficult to understand why this focus was decided upon and how it is intended to inform the mental health field.

The Introduction section does not state the purpose clearly, and provides information about the Covid pandemic and response from 2021 without clarifying that this is the case. For example, the last sentence of the first para states that the impact of C-19 'is expected to be huge and devastating in the coming years', but the reference provided (the WHO covid dashboard) does not support this statement in any way. The measures described as being introduced by the government to manage the Covid-19 situation were similarly from early on in the pandemic, and no information is provided about the situation at the time of data collection. Therefore, the study is not adequately situated in the relevant literature, and the purpose of the study is not made clear.

This weakness continues in the Discussion section. Whilst each individual finding is discussed in relation to relevant literature, the overall purpose and learnings from the study are unclear. The section on 'policy practice and implementation' could usefully be expanded upon.

The data collection, analysis and reporting appears adequate (although I'm not in a position to assess the accuracy of the sampling calculations or the regression analysis). However, the claims made for the meaning of the findings go far beyond what can be concluded from the information available (e.g. that the coping mechanisms identified could be related to challenging work environments, limited access to medications, under-reporting of mental health issues, economic impact of lockdown - there is no evidence to suggest that it is due to any of these factors).

The paper has some potential but would need to be significantly revised to focus more tightly on a clear set of aims and to ground conclusions more solidly in the data.

6. PLOS authors have the option to publish the peer review history of their article (what does this mean?). If published, this will include your full peer review and any attached files.

**Do you want your identity to be public for this peer review?** For information about this choice, including consent withdrawal, please see our Privacy Policy.

Reviewer #1: No

Reviewer #2: No

---

## [Decision Letter · Decision Letter 1]

13 Dec 2024

Psychological distress and coping mechanisms due to the COVID-19 pandemic among the adult population in Bo district Sierra Leone:  cross-sectional study

PMEN-D-24-00222R1

Dear Dr Osborne,

We are pleased to inform you that your manuscript 'Psychological distress and coping mechanisms due to the COVID-19 pandemic among the adult population in Bo district Sierra Leone:  cross-sectional study' has been provisionally accepted for publication in PLOS Mental Health.

Best regards,

Kizito Omona, PhD

Academic Editor

PLOS Mental Health

Reviewer Comments (if any, and for reference):

Reviewer's Responses to Questions

**Comments to the Author**

1. If the authors have adequately addressed your comments raised in a previous round of review and you feel that this manuscript is now acceptable for publication, you may indicate that here to bypass the “Comments to the Author” section, enter your conflict of interest statement in the “Confidential to Editor” section, and submit your "Accept" recommendation.

Reviewer #1: All comments have been addressed

Reviewer #3: All comments have been addressed

2. Does this manuscript meet PLOS Mental Health’s publication criteria? Is the manuscript technically sound, and do the data support the conclusions? The manuscript must describe methodologically and ethically rigorous research with conclusions that are appropriately drawn based on the data presented.

Reviewer #1: Yes

Reviewer #3: Yes

3. Has the statistical analysis been performed appropriately and rigorously?

Reviewer #1: Yes

Reviewer #3: Yes

4. Have the authors made all data underlying the findings in their manuscript fully available (please refer to the Data Availability Statement at the start of the manuscript PDF file)?

Reviewer #1: Yes

Reviewer #3: Yes

5. Is the manuscript presented in an intelligible fashion and written in standard English?

Reviewer #1: No

Reviewer #3: Yes

6. Review Comments to the Author

Reviewer #1: While all the comments were addressed, I would suggest checking the grammar and clarity of the text that was added in response to the reviewers' comments.

Reviewer #3: All comments have been addressed.

7. PLOS authors have the option to publish the peer review history of their article (what does this mean?). If published, this will include your full peer review and any attached files.

**Do you want your identity to be public for this peer review?** For information about this choice, including consent withdrawal, please see our Privacy Policy.

Reviewer #1: No

Reviewer #3: No
